# Establishment of a Virus-Induced Gene Silencing System in *Abelmoschus manihot* L.

**DOI:** 10.3390/plants14020150

**Published:** 2025-01-07

**Authors:** Ting Zhang, Jiaqi Hou, Hongtao Chu, Pengyu Guo, Qianzi Sang, Zhongxu Liu, Li Cao

**Affiliations:** 1Agricultural College, Yanbian University, Yanji 133002, China; zhang_ting0914@163.com (T.Z.); kuromi457@163.com (J.H.); chuhongtao2001@163.com (H.C.); sangqianzi200004@163.com (Q.S.); 19997199021@163.com (Z.L.); 2Laboratory of Molecular Biology of Tomato, Bioengineering College, Chongqing University, Chongqing 400044, China; guopengyucqu@163.com

**Keywords:** *Abelmoschus manihot* L., virus-induced gene silencing, tobacco rattle virus, *AmPDS*, back-of-blade injection

## Abstract

*Abelmoschus manihot* L. (Jinhuakui, JHK) is widely cultivated for its pharmacological properties owing to its high flavonoid content and is commonly used as a garden landscape plant. However, the absence of an efficient genetic transformation system poses significant challenges for functional gene studies in this species. Virus-induced gene silencing (VIGS) is a well-established technique for exploring plant gene functions; however, this technique has not been applied to JHK. Here, a tobacco rattle virus (TRV)–VIGS system was successfully developed for the first time in JHK using the gene encoding phytoene desaturase (*AmPDS*) as a marker gene. This study investigated the impact of various *Agrobacterium* infection methods on the efficiency of *AmPDS* silencing. The results demonstrated that administering two injections—the first on the day of complete cotyledon expansion and the second 14 days later—using pTRV1 and pTRV2–*AmPDS* cultures resuspended to an OD_600_ of 1.0 and via the backside of the blade—led to significant photobleaching in the cotyledons 2 days after the second injection. Subsequent analyses revealed a marked reduction in both chlorophyll content and *AmPDS* expression. These findings suggest that a VIGS system was successfully developed in JHK, thus providing a rapid and effective method for studying gene function in this species and facilitating future research in JHK genetics.

## 1. Introduction

*Abelmoschus manihot* L., an endangered plant from the Malvaceae family and the *Abelmoschus* genus, is recorded as “Jinhuakui” (JHK) in Chinese and is distributed in the Hebei province of China. It is an annual herbaceous plant that is also referred to as wild hibiscus, vegetable hibiscus, sticky dry herb, or mountain elm bark; it is typically sown in the spring and harvested in the fall, with propagation occurring through seeds [1]. Research has shown that the entire JHK plant possesses a wide range of pharmacological properties [2,3], including the alleviation of pain and reduction of inflammation, and is frequently used in the treatment of wounds or sprains. Furthermore, in clinical trials targeting specific diseases, JHK extracts have shown significant therapeutic effects, with notable antioxidant, anticonvulsant, anti-inflammatory, and immunomodulatory activities [4]. The flowers of JHK are particularly rich in trace elements, such as manganese, iron, and magnesium, as well as essential nutrients, including ash, protein, and crude fat [3,5,6,7,8,9]. According to the China–Japan Food Research Center, the seed oil of JHK contains high vitamin E content and six types of fatty acids, with unsaturated fatty acids comprising 68.9% of the total content, which is highly beneficial to human health [9]. Among them, flavonoids serve as the key bioactive constituents, with the highest concentration detected in the flowers (reaching up to 8.47%), followed by the leaves, seeds, stems, and roots [4,10,11,12]. The total flavonoid content in JHK is 10 times higher than that observed in other flavonoid-rich plants, rendering JHK a prime candidate for industrial-scale flavonoid extraction [4,10]. Therefore, comprehensive studies of the active chemical constituents, flavonoid classification, and synthesis mechanisms of JHK are significant in advancing human health.

Virus-induced gene silencing (VIGS) is a technique that is rooted in post-transcriptional gene silencing. The post-transcriptional gene silencing process involves three main stages: initiation, maintenance, and signal amplification and dissemination [13,14]. During the initiation stage, plants recognize and degrade the double-stranded RNA (dsRNA) of the target gene through a post-transcriptional RNA silencing mechanism, producing a significant quantity of siRNA [15]. In the maintenance stage, siRNA facilitates the formation of the RNA-induced silencing complex; the complex can be activated by unwound siRNAs, identify single-stranded siRNA sequences, and degrade complementary transcriptional products [16]. Finally, during the signal amplification and dissemination stage, the siRNA, directed by RNA-dependent RNA polymerase, uses single-stranded RNA as a template to synthesize dsRNA [17,18]. The re-synthesis of dsRNA provides substrates for the generation of additional siRNA, further amplifying the RNA silencing signal and propagating the silencing response across the plant [16,19]. Compared with other transgenic techniques, VIGS presents distinct advantages. First, VIGS is technically straightforward, requiring only *Agrobacterium* infiltration to induce gene silencing without the need for complex genetic transformation steps, thereby allowing the analysis of gene function with greater speed and simplicity, with visible effects typically observed within 2–3 weeks. Furthermore, VIGS is applicable to the analysis of a broad range of plant species, thus allowing for the study of any gene involved in plant growth and development. By designing appropriate target gene fragments, one or more genes can be specifically silenced, thereby addressing gene redundancy issues within gene families and offering greater flexibility in plant gene function research.

The VIGS technique has broad applicability. Currently, viral vectors are available for both monocotyledonous and dicotyledonous plants, thereby facilitating gene function studies across a variety of plant species. For instance, tobacco rattle virus (TRV) has been successfully employed in plants such as the tomato [20], pepper [21], and petunia [22], whereas barley stripe mosaic virus has been applied effectively in monocots such as barley [23]. In addition to RNA viruses, DNA viruses such as geminiviruses, which have been used in *Arabidopsis*, have also shown notable success [24]. Although most viruses face difficulties in infecting plant meristems, which limits the use of viral vectors in certain plants, TRV can efficiently infect plant meristem tissues. Therefore, the TRV-based VIGS system is currently the most widely used system in this context because of its broad host range, high silencing efficiency, and prolonged duration of action.

Phytoene desaturase (PDS) is a key enzyme in the carotenoid biosynthesis pathway because it catalyzes the formation of carotenoid precursors and serves as a critical branch point in the pathway. In turn, PDS silencing disrupts the carotenoid biosynthesis process, leading to the photooxidative-mediated degradation of chlorophyll and, ultimately, plant bleaching [16,17]. Therefore, *PDS* has been widely utilized as a reporter gene in VIGS systems. Currently, limited research exists on gene functions in JHK. In this study, we selected *A. manihot PDS* (*AmPDS*) as a model gene, cloned it, and analyzed its evolutionary relationships using bioinformatics. We also initially established a VIGS system utilizing different *Agrobacterium* inoculation methods. This research provides a foundation for future exploration of key gene functions in JHK.

## 2. Results

### 2.1. Cloning Results of the AmPDS Fragment

In this study, *AmPDS* was initially amplified via polymerase chain reaction (PCR). The PCR product was then analyzed using agarose gel electrophoresis, yielding a single target band of 271 bp (Figure 1), which was consistent with the expected size. The PCR product was then recovered, ligated into the pCE2-TA/Blunt-Zero vector, and transformed into *Escherichia coli* DH5α-competent cells. Positive clones were selected and confirmed via sequencing.

### 2.2. Construction of the AmPDS VIGS Recombinant Vector

After double-enzyme digestion, the target fragment was ligated into the vector and transformed into *E. coli* DH5α-competent cells. Positive clones were selected and confirmed via sequencing, after which the construct was transformed into *Agrobacterium tumefaciens*. The constructed pTRV–*AmPDS* vector is depicted in Figure 2.

### 2.3. Comparison of Silencing Efficiency of Different Inoculation Methods Mediated by Agrobacterium

#### 2.3.1. Back-of-Blade Injection

To detect whether the TRV system was successfully transformed into JHK and enabled to induce the silencing of *AmPDS*, we employed physiological and molecular approaches to observe the phenotype of JHK exhibited in the group injected with the TRV system, pTRV1 + pTRV2 and pTRV1 + pTRV2 − *AmPDS*, and CK group without injection. Firstly, we collected the samples from different groups, and the PCR assay was employed to detect the mRNA that encoded CP protein, a symbolical product from the TRV2 vector. The results showed that there was a clear band in the pTRV1 + pTRV2 and pTRV1 + pTRV2 − *AmPDS* groups while no band in the CK group (Figure 3A), illustrating that the TRV system was successfully transformed into JHK via back-of-blade injection method. Further, the photobleaching phenomenon was observed that two days after the second injection, the cotyledons of JHK seedlings began to show photobleaching, whereas the plants with pTRV1 + pTRV2 exhibited no phenotypic changes, and the true leaves of JHK seedlings infected with pTRV1 + pTRV2 − *AmPDS Agrobacterium* were restored to green after 30 days (Figure 3B). We observed and counted the true leaves of JHK seedlings injected with pTRV1 + pTRV2 − *AmPDS*; the results show that 54.4% of JHK seedlings exhibited photobleaching phenotype in their true leaves (Figure 3C). After the detection of viral infection in JHK leaves, we next analyzed gene expression after infection using qRT-PCR, and the results indicated that *AmPDS* expression in JHK true leaves with phenotypes was significantly reduced compared with the plants injected with pTRV1 + pTRV2, and *AmPDS* expression was reduced by nearly 60% (Figure 3D). Moreover, *AmPDS* silencing led to a significant reduction of the chlorophyll content in JHK leaves with phenotypes (Figure 3E).

#### 2.3.2. Vacuum Infiltration Method

We treated the JHK leaves using the vacuum infiltration method. Firstly, to test whether the TRV-based vector can effectively induce *AmPDS* silencing in JHK, the TRV CP mRNA in the treated JHK leaves was detected by PCR, the results showed that no band was seen in CK, whereas pTRV1 + pTRV2 and pTRV1 + pTRV2 − *AmPDS* had a clear band (Figure 4A), thus indicating that the vacuum infiltration method was successful in achieving infection of JHK leaves. Then, we observed the phenotypes of the treated leaves, and observations showed that 2 days after vacuum infiltration, JHK leaves began to exhibit photobleaching, the incidence of which was 49.2% (Figure 4B,C). To further verify the silencing efficiency of *AmPDS*, the mRNA level of *AmPDS* in JHK leaves with phenotypes was measured using qRT-PCR. The results indicated that compared with the control, this method downregulated *AmPDS* expression in JHK leaves with phenotypes by nearly 30% (Figure 4D). Moreover, *AmPDS* silencing significantly reduced the chlorophyll content in the leaves (Figure 4E).

#### 2.3.3. Direct Soaking Method

The direct soaking method is similar to the vacuum infiltration method. As shown in Figure 5A, no band was seen in CK, but pTRV1 + pTRV2 and pTRV1 + pTRV2 − *AmPDS* had a clear band, which indicated that the direct soaking method was successful in achieving infection of JHK leaves. Two days after direct soaking, JHK leaves began to exhibit photobleaching, the incidence of which was 50.8% (Figure 5B,C). qRT-PCR results indicated that this method downregulated *AmPDS* by nearly 30% (Figure 5D). *AmPDS* silencing significantly reduced the chlorophyll content in the leaves (Figure 5E).

#### 2.3.4. Root Drenching Method

JHK seedlings were inoculated with the pTRV2–*AmPDS* vector via *Agrobacterium* root drenching, and TRV CP mRNA was detected successfully in the treated JHK seedlings (Figure 6A). Observations showed that JHK leaves exhibiting photobleaching phenomena began to appear 7 days after root drenching, and the incidence of photobleaching was 44.4% (Figure 6B,C). The results of qRT-PCR indicated that this method downregulated *AmPDS* by nearly 40% in JHK leaves with phenotypes (Figure 6D). Moreover, *AmPDS* silencing significantly reduced the chlorophyll content in the leaves (Figure 6E).

### 2.4. Comparison of Different Agrobacterium Inoculation Methods

We successfully established a preliminary TRV–VIGS system for JHK, achieving effective TRV-mediated gene silencing, and observed the associated leaf phenotypes. All inoculation methods induced chlorosis and bleaching in the leaves. Among the four methods tested, the back-of-blade injection method resulted in the most significant downregulation of *AmPDS* expression (~60%), thus outperforming the other methods. Therefore, this VIGS system provides a suitable platform for functional studies of genes in JHK seedlings.

## 3. Discussion

PDS plays a crucial role in chlorophyll synthesis in plants, and the silencing of the gene encoding this enzyme results in a prominent photobleaching phenotype. Therefore, the *PDS* gene is often used as a reporter to validate the effectiveness of gene silencing methods. Because of the lack of a well-established silencing system in JHK, we selected *AmPDS* as a reporter gene to develop a VIGS system. To achieve this, we employed various silencing methods, including back-of-blade injection, root drenching, vacuum infiltration, and direct soaking, to identify the VIGS system that is most suitable for JHK seedlings.

Leaf injection is a method that is commonly used for gene silencing in plants because it is particularly suitable for studying genes related to leaves or flowers and has a high silencing efficiency. Ratcliff et al. first applied this method to VIGS, successfully silencing the *NbPDS* gene in tobacco by injecting leaves with *Agrobacterium* cultures carrying TRV [25]. Liu et al. later silenced the *PDS* gene in tomatoes using leaf injection, achieving a silencing efficiency of 50% [26]. Since then, this method has been widely used for gene function analysis in dicotyledonous plants, such as *Arabidopsis* [27], tomato [28], pepper [29], and cotton [30].

Root drenching is primarily used to study genes involved in early root development. Ryu et al. (2004) successfully silenced *NbPDS* in *Nicotiana benthamiana* by applying *Agrobacterium* cultures containing TRV to the root crown area of seedlings [31]. This method has been shown to be effective during the seedling stage in various Solanaceous plants and is more suitable than leaf infiltration methods for studying root-related gene functions. The success rate of this method is nearly 100% in tobacco and ranges from 60% to 70% in tomato, petunia, potato, pepper, and eggplant.

Vacuum infiltration is a time-saving inoculation method that can be applied during the seed stage of plants, expanding the range of genes that can be targeted for VIGS studies, with a success rate of 90–100% [28,32]. Ekengren et al. (2003) successfully silenced *NPR1* and *TGA* in tomatoes by placing seedlings in a vacuum chamber after immersing them in *Agrobacterium* cultures [28]. Hileman et al. (2005) applied vacuum infiltration by immersing poppy seedlings in *Agrobacterium* cultures, successfully obtaining *PapsPDS*-silenced plants [32]. Zhang et al. (2017) used vacuum infiltration to treat germinating wheat and maize seeds with *Agrobacterium* suspensions containing recombinant TRV vectors, successfully producing *PDS*-silenced wheat and maize plants [33].

Under optimal growth conditions, the JHK seedlings began to exhibit the photobleaching phenotype caused by *PDS* silencing within 2–7 days after infection. The measurement of chlorophyll content in the leaves revealed that the levels of chlorophyll a, chlorophyll b, and total chlorophyll in the silenced plants were significantly lower than those detected in the control group. qRT-PCR analysis further confirmed a significant decrease in PDS expression in the silenced JHK plants. These results indicate the successful establishment of a VIGS system in JHK seedlings. By comparing the silencing efficiency of the different methods, we found that the back-of-blade injection method was more effective than the remaining three methods because it downregulated *AmPDS* expression by ~60%. Therefore, the back-of-blade injection method is recommended for gene function validation in JHK.

In addition to VIGS technology, several new gene function identification techniques have emerged in recent years, including the CRISPR/Cas9 gene-editing technology [34]. CRISPR/Cas9 offers high efficiency and precision, enabling precise genome editing, such as gene knockout and modification. This enables the direct investigation of plant gene functions, thereby accelerating the progress of studies of plant gene functions. With the continuous advancement of molecular biotechnology, it is anticipated that more efficient and high-throughput methods of identification of gene functions will be developed in the future. These advancements will provide a deeper understanding of gene functions and regulatory mechanisms in plants, offering strong support to plant breeding and agricultural production.

## 4. Materials and Methods

### 4.1. Plant Materials, Strains, and Plasmids

JHK seeds were collected from Shimen Town, Antu County, Yanbian Korean Autonomous Prefecture, Jilin Province (129.02° E, 43.03° N), and cultivated in a greenhouse at the Beijing Academy of Agriculture and Forestry Sciences. After soaking overnight, the surface water of the seeds was absorbed with gauze, and then the seeds were sown in the soil at 28 °C (16 h light)/18 °C (8 h dark) growing condition. The JHK plants were grown normally as the test material when the cotyledon was fully developed for infection. The collected samples were placed in Ziploc storage bags, flash-frozen in liquid nitrogen, and stored at −80 °C for future use. *E. coli* DH5α-competent cells were purchased from Beijing Qingke Biotechnology Co., Ltd. (Beijing, China), and *A. tumefaciens* EHA105 was obtained from Beijing Huayueyang Biotechnology Co., Ltd. (Beijing, China). The viral vectors pTRV1 and pTRV2 were maintained in our laboratory.

### 4.2. Reagents

The restriction enzymes FastDigest *Eco*RI and *Bam*HI and LB medium (1 L) containing 5 g of yeast extract, 10 g of NaCl, and 10 g of tryptone were weighed using an electronic balance, and distilled water was added to a total volume of 1000 mL. To prepare LB solid medium, 15 g of agar powder was added to the solution, followed by autoclaving at 121 °C for 20 min.

### 4.3. Cloning of the Core Fragment of AmPDS

Total RNA was extracted from JHK leaves using the Vazyme Total RNA Extraction Kit, and 1 µg of RNA was used as a template for reverse transcription, which was performed using the HiScript III 1st Strand cDNA Synthesis Kit (+gDNA wiper) (R312) (Vazyme Biotech, Nanjing, China). The resulting cDNA was diluted for subsequent use. The *AmPDS* gene was identified from a previously generated JHK transcriptome by our research group, and a 299-bp specific fragment within its ORF region was selected based on the SGN website (http://vigs.solgenomics.net/ (accessed on 24 August 2023)). The primers used to clone the gene fragment are listed in Table 1; the pTRV2 vector homologous arm sequences were added at the 5′ ends of both forward and reverse primers. Specific amplification was carried out using the 2× Phanta Flash Master Mix (Dye Plus) (P520) from Vazyme Biotech according to the manufacturer’s instructions. The correct bands in agarose gel electrophoresis were identified, excised, and purified using a gel-extraction technique.

### 4.4. Construction of the Recombinant Viral Vector pTRV–AmPDS

The purified PCR-amplified gene fragment was ligated into the *Eco*RI–*Bam*HI-digested linearized pTRV2 plasmid using SoSoo ligase (TSINGKE TSV-S1 Trelief^®^ SoSoo Cloning Kit from Beijing Qingke Biotechnology Co., Ltd.). The ligation product was then transformed into *E. coli* DH5α-competent cells and positive clones were subsequently identified and sequenced.

### 4.5. Preparation of the Inoculation Solution

*A. tumefaciens* strains containing TRV1, TRV2, and TRV2–*AmPDS* vectors were streaked on LB solid medium containing 100 μg/mL kanamycin and 50 μg/mL rifampicin and incubated at 28 °C for 2 days. Single colonies were picked and cultured overnight at 28 °C and 200 rpm in 3.00 mL of liquid LB medium containing 100 μg/mL kanamycin and 50 μg/mL rifampicin. The cultures were centrifuged at 7000 rpm for 6 min, the supernatant was discarded, and the bacteria were resuspended with the infiltration buffer (10 mM MES, 10 mM MgCl_2_, and 200 µM AS), with the pH adjusted to 5.6–5.8. The final value of OD_600_ was adjusted to 1, and the sample was kept at room temperature for 2–3 h. For infection, the bacteria solution containing pTRV1 and bacteria solution containing pTRV2–*AmPDS* were mixed at a volume ratio of 1:1.

### 4.6. Inoculation Methods

#### 4.6.1. Back-of-Blade Injection Method

JHK seedlings at the dicotyledonous stage were used for gene silencing using the back-of-blade injection method. A 1 mL disposable syringe without a needle was used to aspirate the mixed bacteria solution. The mixed bacteria solution was gently injected into the back of the cotyledon, avoiding the leaf veins until it diffused throughout the blade. Any remaining suspension on the blade surface was wiped off (Figure 7A). The injection was performed twice: once on the day of full cotyledon expansion and again 14 days later. A total of 30 JHK seedlings were injected, and the seedlings were kept in the dark at 25 °C for 24 h, followed by exposure to a 16/8 h light/dark photoperiod. Leaf chlorosis was observed after the second day after the second injection, and 15 JHK true leaves that exhibited significant differences compared with the control group were randomly cut from the petiole. The collected leaves were immediately frozen in liquid nitrogen and stored at −80 °C for PCR and qRT-PCR experiments.

#### 4.6.2. The Method of Vacuum Infiltration

JHK seedlings at the dicotyledonous stage were used, and the leaves were punched into uniform circular disks (avoiding the main veins) using a sterilized hole punch. The cleaned and dried leaf disks were placed in a sterile Petri dish, and the mixed bacteria solution was added until the disks were fully submerged. The Petri dish containing the suspension and leaf disks was placed in a vacuum pump at 0.08 MPa for 2 min, followed by the slow release of the vacuum (Figure 7B); 40 circular disks were treated. The treated leaf disks were then placed on 0.4% water agar and incubated in the dark at 25 °C for 24 h, followed by exposure to a 16/8 h light/dark photoperiod. Leaf chlorosis was observed after the second day, and 20 JHK leaves that exhibited significant differences compared with the control group were randomly selected. The collected leaves were immediately frozen in liquid nitrogen and stored at −80 °C for PCR and qRT-PCR experiments.

#### 4.6.3. The Method of Direct Soaking

JHK seedlings at the dicotyledonous stage were used, and the leaves were punched into uniform circular disks (avoiding the main veins) using a sterilized hole punch. The Petri dish containing the mixed bacteria solution and leaf disks was allowed to stand at room temperature for 20 min (Figure 7C); 40 circular disks were treated. The treated leaf disks were then placed on 0.4% water agar and incubated in the dark at 25 °C for 24 h, followed by exposure to a 16/8 h light/dark photoperiod. Leaf chlorosis was observed after the second day, and 20 JHK leaves that exhibited significant differences compared with the control group were randomly selected. The collected leaves were immediately frozen in liquid nitrogen and stored at −80 °C for PCR and qRT-PCR experiments.

#### 4.6.4. The Method of Root Drenching

JHK seedlings at the dicotyledonous stage were used, with each planting pot containing only a single plant. A sterile pipette tip was used to draw 5 mL of the mixed bacteria solution, which was then applied to the soil near the root zone for drenching (Figure 7D). This inoculation technique was repeated every 7 days for a total of three inoculations, with 30 JHK seedlings treated. On the 7th day after the third treatment, the leaves began to show chlorosis. 15 JHK leaves that exhibited significant differences compared with the control group were randomly cut from the petiole, immediately frozen in liquid nitrogen, and stored at −80 °C for PCR and qRT-PCR experiments.

### 4.7. Gene Expression Analysis

Total RNA was extracted from treated JHK leaves using the Vazyme Total RNA Extraction Kit, and cDNA was synthesized using the HiScript III All-in-one RT SuperMix Perfect for qPCR (R333) from Vazyme Biotech. Quantitative analysis was carried out on a real-time PCR platform. The primer sequences used are listed in Table 2. The PCR conditions were as follows: 95 °C for 30 s, followed by 40 cycles of 95 °C for 10 s and 60 °C for 30 s, with a final melting curve analysis. Actin was used as the internal reference gene. The 2^−ΔΔCᴛ^ method was employed to analyze qRT-PCR expression data, according to Zhang et al. [35]. Three biological replicates were performed for all qRT-PCR assays. Significant differences in the development assay were determined using Student’s *t*-tests (* *p* < 0.05; ** *p* < 0.01). The quantitative analysis was performed using GraphPad Prism (v.9.0.0).

### 4.8. TRV CP mRNA Detection

Total RNA was extracted from treated JHK leaves using the Vazyme Total RNA Extraction Kit, and the reagents listed in Table 3 were used to synthesize cDNA. The components were gently mixed with a pipette, followed by brief centrifugation to remove any bubbles. This was followed by incubation at 37 °C for 1 h by PCR and storage at −20 °C for future use. Using cDNA as a template, primers (listed in Table 4), ddH_2_O, and enzyme mix (2× Rapid Taq Master Mix (P222) from Vazyme Biotech) were added to conduct the reaction of specifically amplifying TRV CP mRNA.

### 4.9. Determination of Chlorophyll Content

Chlorophyll content was measured using a Plant Chlorophyll Content Assay Kit (BC0990, Solarbio, Beijing, China). The detection assays were conducted using a 0.1-g sample, and three biological replicates were analyzed. Significant differences were determined using Student’s *t*-tests (* *p* < 0.05; ** *p* < 0.01). Quantitative analysis was performed using GraphPad Prism (v.9.0.0).

### 4.10. Statistical Analyses

Data were analyzed with Excel 2021 software (Microsoft Corp., Redmond, WA, USA). All statistical analyses were performed with GraphPad Prism (v.9.0.0). A Student’s *t*-test was used to assess differences between different treatments (* *p* < 0.05, ** *p* < 0.01).

## Figures and Tables

**Figure 1 plants-14-00150-f001:**
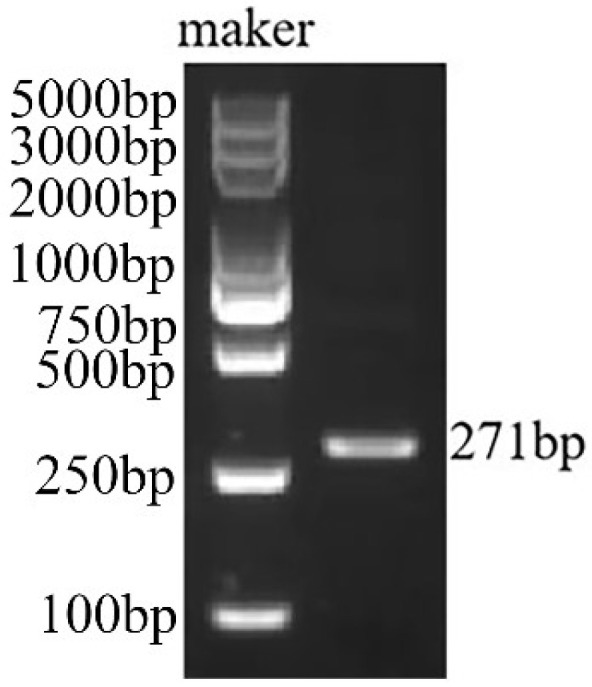
Electrophoresis of the amplified fragment of *AmPDS*.

**Figure 2 plants-14-00150-f002:**
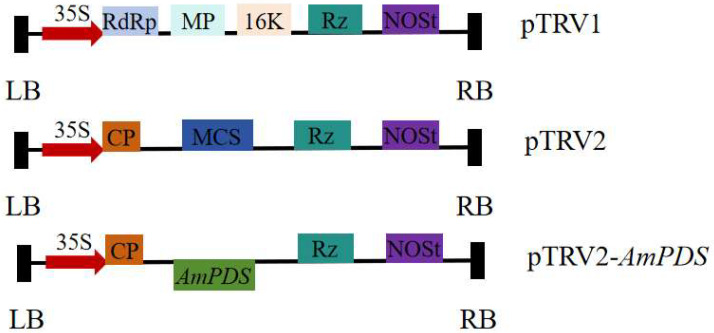
Construction of the *AmPDS* VIGS vector. LB and RB—left and right borders of T-DNA, respectively; 35S—duplicated CaMV 35S promoter; RdRp—RNA-dependent RNA polymerase; MP—movement protein; CP—coat protein; 16K—16 kDa cysteine-rich protein; MCS—multiple cloning sites; Rz—self-cleaving ribozyme; NOSt—nopaline synthase terminator.

**Figure 3 plants-14-00150-f003:**
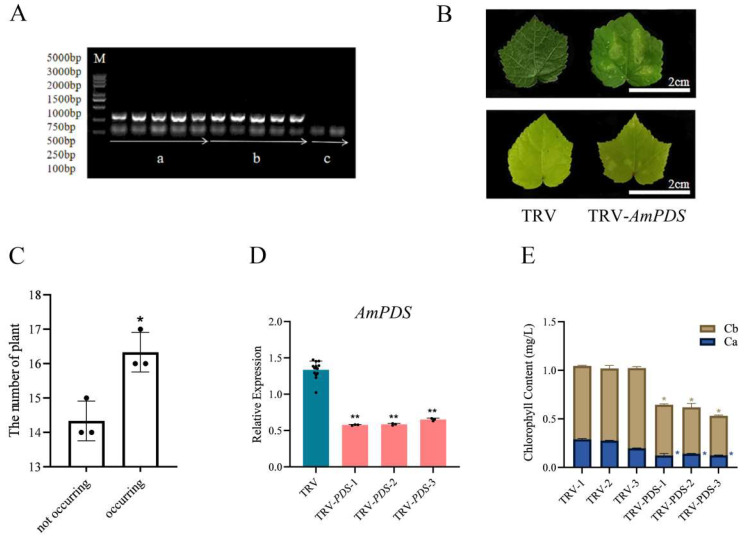
The evaluation of the back-of-blade injection method for the silencing of *AmPDS*. (**A**) The detection of mRNA-encoded CP protein using PCR assay in the different groups; M—DNA marker; a—PCR product from samples injected with pTRV1 + pTRV2; b—PCR product from samples injected with pTRV1 + pTRV2 − *AmPDS*; c—PCR product from samples without injection. (**B**) Phenotype of JHK leaves respectively injected with pTRV1 + pTRV2 and pTRV1 + pTRV2 − *AmPDS*; top panel, 2 days after injection; bottom panel, 30 days after injection. (**C**) The number of plants in which photobleaching phenotype occurred and did not occur after injection with pTRV1 + pTRV2 − *AmPDS*. The value means average ± bar and * means *p* < 0.05 by *t*-test. (**D**) The detection of the transcript level of *AmPDS*, using qRT-PCR assay, in plants with phenotypes injected with pTRV1 + pTRV2 − *AmPDS*, compared with plants injected with pTRV1 + pTRV2 only. Three JHK leaves with phenotypes were selected as one biological replicate in this experiment, and the three independent biological repeats were set. The value means average ± bar, and ** means *p* < 0.01 by *t*-test. (**E**) The chlorophyll content was examined from samples that exhibited phenotypes and samples injected with pTRV1 + pTRV2 only (The blue * represents the significance analysis results of chlorophyll a, and the orange * represents the significance analysis results of chlorophyll b). Three JHK leaves with phenotypes were selected as one biological replicate in this experiment, and the three independent biological repeats were set. The value means average ± bar and * means *p* < 0.05 by *t*-test. All statistical analyses were performed with GraphPad Prism (v.9.0.0). A Student’s *t*-test was used to assess differences between different treatments (* *p* < 0.05, ** *p* < 0.01).

**Figure 4 plants-14-00150-f004:**
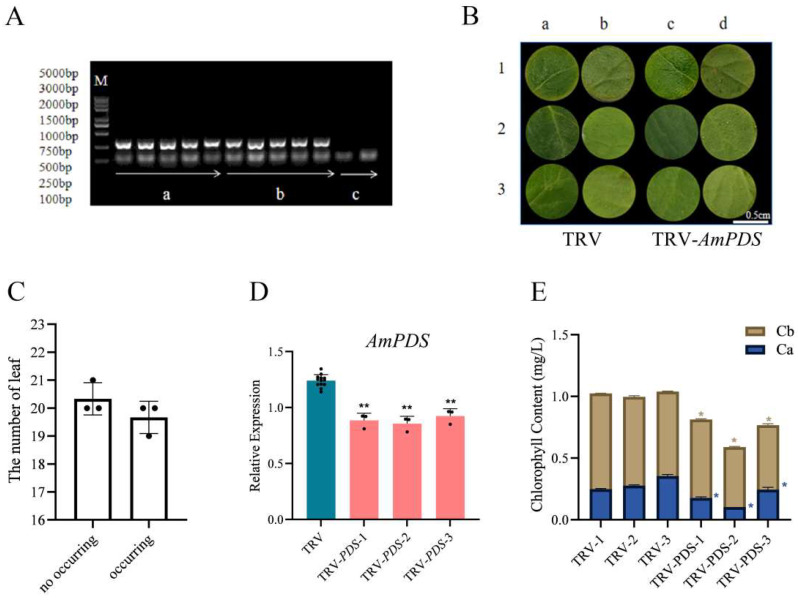
The evaluation of the vacuum infiltration method for the silencing of *AmPDS*. (**A**) The detection of mRNA-encoded CP protein using PCR assay in the different groups; M—DNA marker; a—PCR product from samples vacuum-infiltrated with pTRV1 + pTRV2; b—PCR product from samples vacuum-infiltrated with pTRV1 + pTRV2 − *AmPDS*; c—PCR product from samples without vacuum infiltration. (**B**) Phenotype of JHK leaves vacuum-infiltrated with pTRV1 + pTRV2 and pTRV1 + pTRV2 − *AmPDS*, respectively; ac—adaxial side; bd—back of the blade. (**C**) The number of plants in which photobleaching by phenotype occurred and did not occur after vacuum infiltration with pTRV1 + pTRV2 − *AmPDS*. (**D**) The detection of the transcript level of *AmPDS*, using qRT-PCR assay, in plants with phenotypes vacuum-infiltrated with pTRV1 + pTRV2 − *AmPDS*, compared with plants vacuum-infiltrated with pTRV1 + pTRV2 only. Three JHK leaves with phenotypes were selected as one biological replicate in this experiment, and the three independent biological repeats were set. The value means average ± bar, and ** means *p* < 0.01 by *t*-test. (**E**) The chlorophyll content was examined from samples that exhibited phenotype and samples vacuum-infiltrated with pTRV1 + pTRV2 only (The blue * represents the significance analysis results of chlorophyll a, and the orange * represents the significance analysis results of chlorophyll b). Five JHK leaves with phenotypes were selected as one biological replicate in this experiment, and the three independent biological repeats were set. The value means average ± bar and * means *p* < 0.05 by *t*-test. All statistical analyses were performed with GraphPad Prism (v.9.0.0). A Student’s *t*-test was used to assess differences between different treatments (* *p* < 0.05, ** *p* < 0.01).

**Figure 5 plants-14-00150-f005:**
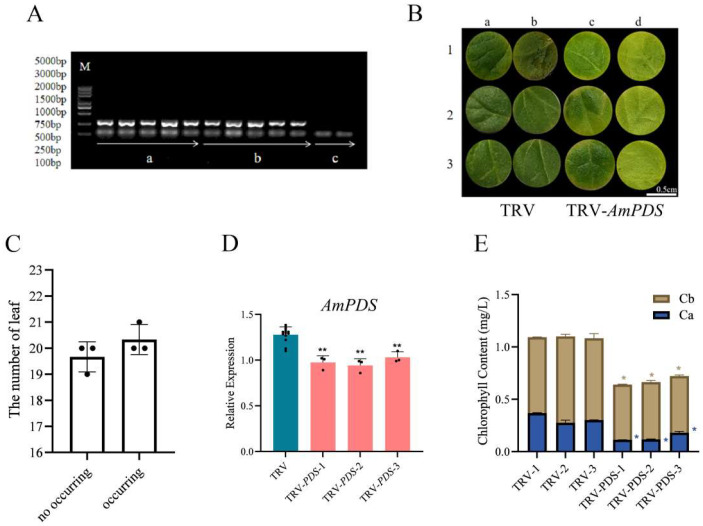
The evaluation of the direct soaking method for the silencing of *AmPDS*. (**A**) The detection of mRNA-encoded CP protein using PCR assay in the different groups; M—DNA marker; a—PCR product from samples directly soaked with pTRV1 + pTRV2; b—PCR product from samples directly soaked with pTRV1 + pTRV2 − *AmPDS*; c—PCR product from samples without direct soaking. (**B**) The phenotype of JHK leaves is directly soaked with pTRV1 + pTRV2 and pTRV1 + pTRV2 − *AmPDS*, respectively; ac—adaxial side; bd—back of the blade. (**C**) The number of plants in which photobleaching phenotype occurred and did not occur after direct soaking with pTRV1 + pTRV2 − *AmPDS*. (**D**) The detection of the transcript level of *AmPDS*, using qRT-PCR assay, in plants with phenotypes directly soaked with pTRV1 + pTRV2 − *AmPDS*, compared with plants directly soaked with pTRV1 + pTRV2 only. Three JHK leaves with phenotypes were selected as one biological replicate in this experiment, and the three independent biological repeats were set. The value means average ± bar, and ** means *p* < 0.01 by *t*-test. (**E**) The chlorophyll content was examined from samples that exhibited phenotypes and samples directly soaked with pTRV1 + pTRV2 only (The blue * represents the significance analysis results of chlorophyll a, and the orange * represents the significance analysis results of chlorophyll b). Five JHK leaves with phenotypes were selected as one biological replicate in this experiment, and the three independent biological repeats were set. The value means average ± bar and * means *p* < 0.05 by *t*-test. All statistical analyses were performed with GraphPad Prism (v.9.0.0). A Student’s *t*-test was used to assess differences between different treatments (* *p* < 0.05, ** *p* < 0.01).

**Figure 6 plants-14-00150-f006:**
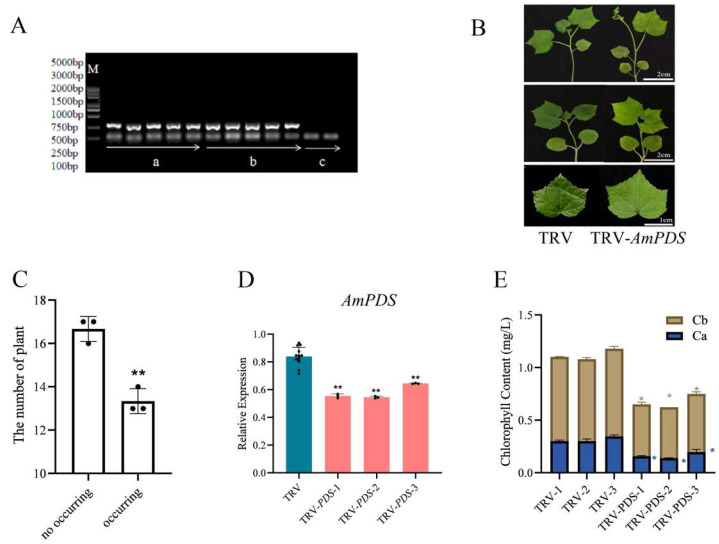
The evaluation of root drenching method for the silencing of *AmPDS*. (**A**) The detection of mRNA-encoded CP protein using PCR assay in different groups; M—DNA marker; a—PCR product from samples root-drenched with pTRV1 + pTRV2; b—PCR product from samples root-drenched with pTRV1 + pTRV2 − *AmPDS*; c—PCR product from samples without root drenching. (**B**) The phenotype of JHK leaves root-drenched with pTRV1 + pTRV2 and pTRV1 + pTRV2 − *AmPDS*, respectively; top panel—7 days after root drenching; middle panel—14 days after root drenching; bottom panel—21 days after root drenching. (**C**) The number of plants in which photobleaching phenotype occurred and did not occur after root drenching with pTRV1 + pTRV2 − *AmPDS*. The value means average ± bar and ** means *p* < 0.01 by *t*-test. (**D**) The detection of the transcript level of *AmPDS*, using qRT-PCR assay, in plants with phenotypes root-drenched with pTRV1 + pTRV2 − *AmPDS*, compared with plants root-drenched with pTRV1 + pTRV2 only. Three JHK leaves with phenotypes were selected as one biological replicate in this experiment, and the three independent biological repeats were set. The value means average ± bar, and ** means *p* < 0.01 by *t*-test. (**E**) The examination of chlorophyll content from samples that exhibited phenotypes and samples root-drenched with pTRV1 + pTRV2 only (The blue * represents the significance analysis results of chlorophyll a, and the orange * represents the significance analysis results of chlorophyll b). Three JHK leaves with phenotypes were selected as one biological replicate in this experiment, and the three independent biological repeats were set. The value means average ± bar and * means *p* < 0.05 by *t*-test. All statistical analyses were performed with GraphPad Prism (v.9.0.0). A Student’s *t*-test was used to assess differences between different treatments (* *p* < 0.05, ** *p* < 0.01).

**Figure 7 plants-14-00150-f007:**
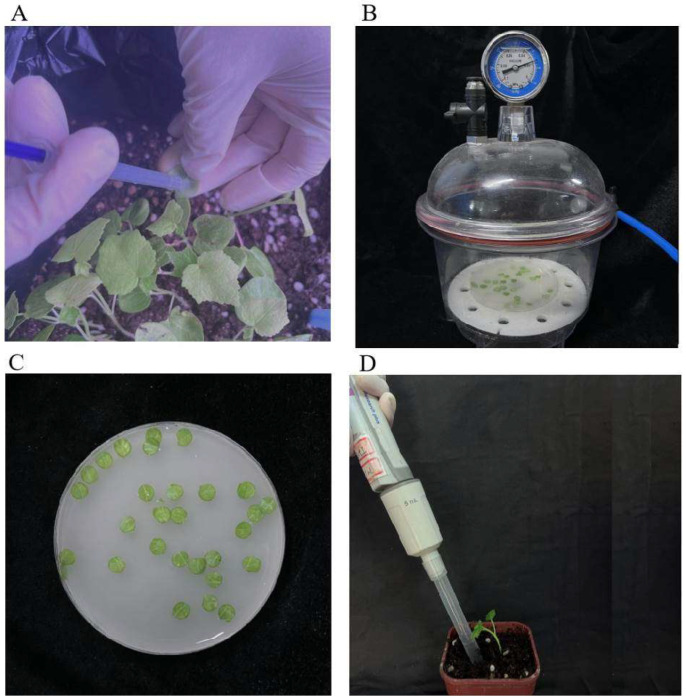
Schematic diagram of the infection method for virus-induced gene silencing. (**A**) Back-of-blade injection method. (**B**) Vacuum infiltration method. (**C**) Direct soaking method. (**D**) Root drenching method.

**Table 1 plants-14-00150-t001:** Primer sequence.

Primer		Sequence (5′ to 3′)
*AmPDS*	Forward	ATGCTGACTTGGCCTGAGAAAG
	Reverse	TCAGTGTCTGCTTGCTCCAGTC
*AmPDS* silencing fragment	Forward	gtgagtaaggttaccgaattcATGCTGACTTGGCCTGAGAAAG
	Reverse	cgtgagctcggtaccggatccCCAAGAATGCCATCTTTGATCC

**Table 2 plants-14-00150-t002:** Primer information in *A. manihot* L.

Number	Gene Name		Sequence (5′ to 3′)
1	*AmPDS*	Forward	CCTGATCGTGTGACTGAGGA
		Reverse	ATTGGCATGCAAAGCCTCTC
2	Actin	Forward	TCTTTCATCGGGATGGAAGC
		Reverse	ACTGAGCACAATGTTACCGTAGAG

**Table 3 plants-14-00150-t003:** Reaction system.

Reagent	Volume
5× M-MLV buffer	2 μL
dNTP Mix (2.5 mM)	2 μL
Random primer (10 μM)	0.5 μL
Oligo(dT)_18_ (10 μM)	0.5 μL
RNase inhibitor (40 U/μL)	0.25 μL
M-MLV reverse transcriptase (200 U/μL)	0.25 μL
RNA	1 μL
RNase-free ddH_2_O	To a final volume of 10 µL

**Table 4 plants-14-00150-t004:** Primer sequences.

Primer		Sequence (5′ to 3′)
TRV CP mRNA	Forward	CCTGCTGACTTGATGGACGA
	Reverse	CCAGTGTTCGCCTTGGTAG

## Data Availability

The datasets analyzed during the current study are available in the NCBI BioProject repository, PRJNA786451. The datasets used and/or analyzed during the current study are available from the corresponding author upon reasonable request due to privacy.

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
