# Peer review of "Establishment of a Virus-Induced Gene Silencing System in Abelmoschus manihot L."

_plants, 2025, doi:10.3390/plants14020150_

Round 1
Reviewer 1 Report
Comments and Suggestions for Authors
This manuscript is the first report on the virus-induced gene silencing in Hibiseu manihot. In this point, the study is novel, and the manuscript can be published in the journal. However, it should be revised and the authors are encouraged to ask a native English speaker to check English on the revised manuscript before resubmission. Some comments on the manuscript is written below. Hopefully, they will contribute to the author's revision of the manuscript.
L36
Malvaceae, the family name of the studied plant, must not be italicised.
L47
The atomic symbol, Mg, should be removed. Otherwise, the atomic symbol of the other metals should be provided.
L97
PDS must not be italicised, since it is the acronym of the enzyme (L95), not gene.
L105
The title of the section should be written as "2. Results" in bold characters.
L123
Agrobacterium, a genus name, should be written in stand character, since the title of this section is written in italics.
L133
What is "empty vector"? Does it pathogen-free vector?
L135
What measurements are referred to?
L135
Provide the results of statistical analyses for the significance.
Figure 3B, C
Provide the unit on the vertical axis.
L153
Give the results of the statistical analyses for the significance.
L162
Give the t-value and df.
L224
Solanaceous should be written in stand characters.
L258
Materials and Methods may be replace as the title of the section.
Describe the statistical analyses clearly and in detail that were used for this study. Since no descriptions on statistical analyses for this study are given in the manuscript, it is impossible to decide whether the study was performed correctly in a scientific sense.
L260-261
Refer to the date when the seeds were collected and describe briefly how the plants were cultivated with the physical conditions and any treatment with chemical substances.
L262
What was sampled? The sampling method should be described.
L306
How many seedlings were collected how and when?
L308
Explain what the bacterial suspensions was.
L315
Refer to how many seedlings were sampled how and when.
L332
What is "bacterial suspension"? Was this different from that on L308?
Comments on the Quality of English LanguageEnglish in the manuscript needs to be edited by a native English speaker.
Author Response
Dear Editor,
We extend our heartfelt appreciation for your valuable time and dedicated effort expended in the review of our manuscript titled " Establishment of a Virus-induced Gene Silencing System in Hibiseu manihot L." (Manuscript ID: plants-3282314). We sincerely appreciate the constructive criticism provided and would like to express our gratitude for the insightful comments made by the reviewers. In accordance with the feedback from the editor and reviewers, we have diligently addressed the suggestions and concerns raised during the review process. We have carefully revised the manuscript, and the changes have been highlighted in red for easy reference in the revised version. Additionally, we have provided detailed responses to each reviewer's comments, which are presented below, along with the corresponding location of the revisions in the manuscript.
Reviewer 1
Comments and Suggestions for Authors
This manuscript is the first report on the virus-induced gene silencing in Hibiseu manihot. In this point, the study is novel, and the manuscript can be published in the journal. However, it should be revised and the authors are encouraged to ask a native English speaker to check English on the revised manuscript before resubmission. Some comments on the manuscript is written below. Hopefully, they will contribute to the author's revision of the manuscript.
- L36
Malvaceae, the family name of the studied plant, must not be italicised.
Response: We were really sorry for our careless mistakes. Thanks for your reminder. In our revised manuscript, we modified Malvaceae, line 40.
L47
The atomic symbol, Mg, should be removed. Otherwise, the atomic symbol of the other metals should be provided.
Response: We sincerely thank the reviewer for careful reading. As suggested by the reviewer, in our revised manuscript, we removed the atomic symbol, Mg, line 49.
L97
PDS must not be italicised, since it is the acronym of the enzyme (L95), not gene.
Response: Thank you for your constructive suggestion, we feel sorry for our carelessness. In our revised manuscript, we have corrected the font formatting of PDS in the revised manuscript as per the reviewer’s suggestion, line 90.
L105
The title of the section should be written as "2. Results" in bold characters.
Response: Thanks for your comments. In our revised manuscript, we modified the title, line 99.
L123
Agrobacterium, a genus name, should be written in stand character, since the title of this section is written in italics.
Response: Thanks for your careful checks. We are sorry for our carelessness. Based on your comments, we modified the Agrobacterium in our entire revised manuscript.
L133
What is "empty vector"? Does it pathogen-free vector?
Response: Thank you for your constructive suggestion. The representation of “empty vector” means only the leaves of JHK that were injected with pTRV1+pTRV2. We correct the inaccurate words and try our best to make a scientific editing, line 130.
L135
What measurements are referred to?
Response: Thanks for your comments. The representation of “measurements” means that chlorophyll content in JHK leaves. We correct the inaccurate words and try our best to make a scientific editing, line 131.
L135
Provide the results of statistical analyses for the significance.
Response: Thanks for the insightful comment. We have provided the results of statistical analyses for the significance for Figure 3E, P value = 0.0132.
Figure 3B, C
Provide the unit on the vertical axis.
Response: Thanks for the insightful comment. We have provided the unit on the vertical axis as Figure 3B. For Figure 3C, the 2−ΔΔCᴛ method was employed to analyze qRT-PCR expression data according to Zhang et al.[33]. Three biological replicates were performed for all qRT-PCR assays. Significant differences in the development assay were determined using Student's t-tests (*p < 0.05; **p < 0.01). The quantitative analysis was performed using the GraphPad Prism (v.9.0.0) software.
L153
Give the results of the statistical analyses for the significance.
Response: Thanks for the insightful comment. We have provided the results of statistical analyses for the significance for Figure 4E, P value = 0.2302.
L162
Give the t-value and df.
Response: We sincerely appreciate your valuable input and suggestion. In our study, t=1.414, df=4 of the Figure 4E.
L224
Solanaceous should be written in stand characters.
Response: We appreciate your attention to detail and valuable feedback. According to your suggestion, in our revised manuscript, we have corrected the font formatting of Solanaceous in the revised manuscript as per the reviewer’s suggestion , line 234.
L258
Materials and Methods may be replace as the title of the section.
Response: Thanks for your comments. In our revised manuscript, we modified the title, line 266.
Describe the statistical analyses clearly and in detail that were used for this study. Since no descriptions on statistical analyses for this study are given in the manuscript, it is impossible to decide whether the study was performed correctly in a scientific sense.
Response: We sincerely thank the reviewer for careful reading. The data were analyzed with Excel 2021 software (Microsoft Corp., Redmond, WA,USA). All statistical analyses were performed with GraphPad Prism (v.9.0.0). Student’s t-test were used to assess differences between different treatments ( *: p < 0.05, **: p < 0.01).
L260-261
Refer to the date when the seeds were collected and describe briefly how the plants were cultivated with the physical conditions and any treatment with chemical substances.
Response: Thanks for your comments. In our revised manuscript, we modified the title, line 268.
In presents study, JHK seeds were collected from Shimen Town, Antu County, Yanbian Korean Autonomous Prefecture, Jilin Province (129.02°E, 43.03°N), and cultivated in a greenhouse at the Beijing Academy of Agriculture and Forestry Sciences. After soaking overnight, the surface water of the seeds was absorbed with gauze, and then the seeds were sown in the soil, 28℃(16 h) light /18℃(8 h) dark was set as the plant growing condition, and the JHK plants growing normally as the test material. The collected samples were placed in ziplock storage bags, flash frozen in liquid nitrogen, and stored at –80°C for future use.
L262
What was sampled? The sampling method should be described.
Response: Thank you for your constructive suggestion. The sample was the leaves of JHK with pTRV1+pTRV2 and pTRV1+pTRV2-AmPDS were collected. Cut the treated leaves from the petioles using scissors, the collected leaves were immediately frozen in liquid nitrogen and stored at −80 ℃, each leaf is an independent sample.
L306
How many seedlings were collected how and when?
Response: Thank you very much for your careful comments. In our revised manuscript, leaves were cut from the petiole and quickly placed into liquid nitrogen and stored at −80 ℃, Leaf chlorosis was observed after the second day after the second injection, and 15 leaves of JHK that were injected with pTRV1+pTRV2 and pTRV1+pTRV2-AmPDS were collected, line 322-326.
L308
Explain what the bacterial suspensions was.
Response: Thanks for your comments. The representation of “bacterial suspensions” means that agrobacterium tumefaciens strains containing TRV1, TRV2, and TRV2–HmPDS vectors were streaked on LB solid medium containing 100 μg/mL kanamycin and 50 μg/mL rifampicin, and incubated at 28°C for 2 days. Single colonies were picked and cultured overnight at 28°C and 200 rpm in 3.00 mL of liquid LB medium containing 100 μg/mL kanamycin and 50 μg/mL rifampicin. The cultures were centrifuged at 7000 rpm for 6 min, and the supernatant was discarded and the bacteria were re-suspended with the infiltration buffer (10 mM MES, 10 mM MgCl2 and 200 µM AS)., with the pH adjusted to 5.6–5.8. The final value of OD600 was adjusted to 1, and stood at room temperature for 2–3 h. . We correct the inaccurate words and try our best to make a scientific editing, line 305 to line 312.
L315
Refer to how many seedlings were sampled how and when.
Response: Thank you very much for your suggestion. In our revised manuscript, leaves were cut from the petiole and quickly placed into liquid nitrogen and stored at −80 ℃, Leaf chlorosis was observed after the second day after the second injection, and 15 leaves of JHK that were injected with pTRV1+pTRV2 and pTRV1+pTRV2-AmPDS were collected.
L332
What is "bacterial suspension"? Was this different from that on L308?
Response: Thank you for your valuable comment. The "bacterial suspension" is the same as the L308. We correct the inaccurate words and try our best to make a scientific editing, line 341.
We tried our best to improve the manuscript and made some changes marked in red in revised paper which will not influence the content and framework of the paper. We appreciate for Editors and Reviewer’ warm work earnestly, and hope the correction will meet with approval.
Finally, we thank again for reviewers’ helpful comments and advices. We hope that the revisions adequately address the concerns raised during the review better meets the standards set by "Plants". We sincerely welcome comments for a further improvement of the paper. Thank you once again for your valuable guidance and support throughout the review process.
Sincerely,
Corresponding author: Li Cao
Affiliation: Agricultural College of Yanbian University
Email Address: lcao@ybu.edu.cn

Reviewer 2 Report
Comments and Suggestions for Authors
This manuscript showed the VIGS in the Hibiseu manihot plant. The provided results may be useful for the specific readers who study the physiology and molecular biology of H. manihot plants. However, the present style have many serious issues that should be improved. So intensive revise is required before publication.
comments (Q means query)
(1) Line 36. Hibiseu manihot belong to the Abelmoschus genus.
(Q) This description confuse the readers. The genus Hibiseu manihot is Hibseu, not Abelmoschus. So some additional explanation should be provided. The authors should clearly show the synonym.
(2) Line 36 to line 60
(Q) In this paragraph, there are several repeated description about the pharmacological effects. It should be summarize in more short style.
(3) Line 62 to line 82
(Q) The references in this paragraph are all too old. They should be updated. Ref 15 showed Drosophila RISC not plant one. It still remains to be solved the molecular weight of plant RNA induced silencing complex (RISC). In addition, siRNA amplification in plant does not require the siRNA primer. Most siRNA amplification occur in a independent manner from the siRNA primer.
(4) Figure 3B, 4B, 5B
(Q) The top of bars is unusual. What is the small bars (I see three mini bars in the top of each bar)?
(5) Figure legends for Fig. 3, 4, 5, and 6. A. CP protein virus detection.
(Q) I cannot understand this sentence. In these figure, the PCR-amplified DNA fragments are shown, and not for CP protein.
(6) Figure legends for Fig. 3, 4, 5, and 6. B. CP protein detection rate.
(Q) The authors did not investigate any protein level of CP protein. They only detected the RNA level of TRV via detection of RNA encoding CP protein.
(7) Text for the description of Fig. 3, 4, 5, and 7.
For example, the authors described the result of Figure 3A,3B as follows; the successful introduction of theTRV vector was confirmed base on the detection of the CP protein.
(Q) As already pointed out, the authors did not determine any protein level of CP protein. They only detected the level of RNA encoding CP protein.
(8) Figure legends for Fig. 3, 4, 5, and 6.
(Q) The details of each experiment were not shown. For example, in Fig. 3, the authors injected Agrobacterium suspension into the cotyledon two times. Then the Fig. 3D, they showed the leaf photograph. Which leaf was investigated? In addition, they mentioned that the TRV-HmPDS leaves showed photobleaching. However, the Fig. 3D (b) showed that the leaves in test sample was more greenish than the control sample. How do you judge the “photobleached” leaves ? Anyway the details of all experiments for Fig. 3, 4, 5 and 7 should be provided in more detail.
(9) Text for the description of Fig. 3, 4, and 5.
For example, the authors described the result of these experiments as follows; JHK seedlings were inoculated via Agrobacterium infiltration. However the method for each experiment was different. So, the authors should provide unique sentences in each experiment. Do not repeat the similar sentences in the text!
(10) Line 246. leaf abaxial injection method
(Q) The authors should unify the words. In line 124 and method paragraph, they use “Back-of blade injection”. However, they use here “leaf abaxial injection method” even though these two showed the same experiment.
(11) 4.6.1 Back-of-blade injection method
(Q) As mentioned in the previous query, authors injected the Agrobacterium solution into the abaxial side of cotyledon (same as back-of blade). However, in the method (4.6.1), authors mentioned that suspention was injected into the leaf side of the cotyledon. Usually, leaf side means the adaxial side and may be not abaxial side. Which was the true?
(12) Figure legends for Fig. 4 and 5.
The authors used “pressure side”
(Q) I cannot understand “pressure side”. It means the adaxial side.
Comments on the Quality of English LanguageThe intense revise is required.
Author Response
Dear Editor,
We extend our heartfelt appreciation for your valuable time and dedicated effort expended in the review of our manuscript titled " Establishment of a Virus-induced Gene Silencing System in Hibiseu manihot L." (Manuscript ID: plants-3282314). We sincerely appreciate the constructive criticism provided and would like to express our gratitude for the insightful comments made by the reviewers. In accordance with the feedback from the editor and reviewers, we have diligently addressed the suggestions and concerns raised during the review process. We have carefully revised the manuscript, and the changes have been highlighted in red for easy reference in the revised version. Additionally, we have provided detailed responses to each reviewer's comments, which are presented below, along with the corresponding location of the revisions in the manuscript.
Reviewer 2
This manuscript showed the VIGS in the Hibiseu manihot plant. The provided results may be useful for the specific readers who study the physiology and molecular biology of H. manihot plants. However, the present style have many serious issues that should be improved. So intensive revise is required before publication.
comments (Q means query)
(1) Line 36. Hibiseu manihot belong to the Abelmoschus genus.
(Q) This description confuse the readers. The genus Hibiseu manihot is Hibseu, not Abelmoschus. So some additional explanation should be provided. The authors should clearly show the synonym.
Response: We sincerely thank the reviewer for careful reading. As suggested by the reviewer, through consulting a large number of literature materials and after careful research and discrimination, we have corrected the “Hibiseu manihot” into “Abelmoschus manihot”. In this way, this accurate description enables readers to understand the relevant information about the plant JHK more clearly and distinctly.
(2) Line 36 to line 60
(Q) In this paragraph, there are several repeated description about the pharmacological effects. It should be summarize in more short style.
Response: We appreciate your attention to detail and valuable feedback. In our revised manuscript, According to your suggestion, we modified the part, line 40 to line 57.
Abelmoschus manihot L., one of the endangered plants from the family of Malvaceae and the genus of Abelmoschus, is recorded as “Jinhuakui” (JHK) in Chinese, and distributed in the Hebei province of China, is an annual herbaceous plant that is also referred to as wild hibiscus, vegetable hibiscus, sticky dry herb, or mountain elm bark; it is typically sown in the spring and harvested in the fall, with propagation occurring through seeds [1]. Research has shown that the entire JHK plant possesses a wide range of pharmacological properties [2,3], including the alleviation of pain and reduction of inflammation, and are frequently used in the treatment of wounds or sprains. Furthermore, in clinical trials targeting specific diseases, JHK extracts have significant therapeutic effects, with notable antioxidant, anticonvulsant, anti-inflammatory, and immunomodulatory activities [4]. The flowers of JHK are particularly rich in trace elements, such as manganese, iron, and magnesium, as well as essential nutrients, including ash, protein, and crude fat [3,5–9]. According to the China–Japan Food Research Center, the seed oil of JHK contains a high vitamin E content and six types of fatty acids, with unsaturated fatty acids comprising 68.9% of the total content, which is highly beneficial to human health [9]. Among them, flavonoids serve as the key bioactive constituents, with the highest concentration detected in the flowers (reaching up to 8.47%), followed by the leaves, seeds, stems, and roots [4,10–12]. The total flavonoid content in JHK is 10 times higher than that observed in other flavonoid-rich plants, rendering it a prime candidate for industrial-scale flavonoid extraction [4,10]. Therefore, comprehensive studies of the active chemical constituents, flavonoid classification, and synthesis mechanisms of JHK are significant in advancing human health.
(3) Line 62 to line 82
(Q) The references in this paragraph are all too old. They should be updated. Ref 15 showed Drosophila RISC not plant one. It still remains to be solved the molecular weight of plant RNA induced silencing complex (RISC). In addition, siRNA amplification in plant does not require the siRNA primer. Most siRNA amplification occur in a independent manner from the siRNA primer.
Response: We appreciate your attention to detail and valuable feedback. In our revised manuscript, According to your suggestion, we modified the part, line 59 to line 77.
Virus-induced gene silencing (VIGS) is a technique that is rooted in post-transcriptional gene silencing (PTGS). The PTGS process involves three main stages: initiation, maintenance, and signal amplification and dissemination [13,14]. During the initiation stage, plants recognize and degrade the dsRNA of the target gene through a post-transcriptional RNA silencing mechanism, leading to the production of a significant quantity of siRNA[15]. In the maintenance stage, siRNA facilitates the formation of the RNA-induced silencing complex, which consists of RNA and proteins, with a molecular weight of approximately 500 kDa [16]. Finally, during the signal amplification and dissemination stage, the siRNA, directed by RNA-dependent RNA polymerase, uses single-stranded RNA as a template to synthesize double-stranded RNA (dsRNA),[17,18]. The re-synthesis of dsRNA provides substrates for the generation of additional siRNA, further amplifying the RNA silencing signal and propagating the silencing response across the plant [19,20]. Compared with other transgenic techniques, VIGS presents distinct advantages. First, VIGS is technically straightforward, requiring only Agrobacterium infiltration to induce gene silencing, without the need for complex genetic transformation steps, thereby allowing the analysis of gene function with greater speed and simplicity. Second, VIGS is highly time efficient, with visible effects typically observed within 2–3 weeks. Moreover, VIGS is broadly applicable, thus allowing the study of any gene involved in plant growth and development. By designing appropriate target gene fragments, one or more genes can be specifically silenced, thereby addressing gene redundancy issues within gene families and offering greater flexibility in plant gene function research.
- Rössner, C.; Lotz, D.; Becker, A. VIGS Goes Viral: How VIGS Transforms Our Understanding of Plant Science. Annu. Rev. Plant Biol.2022, 73, 703–728, doi:10.1146/annurev-arplant-102820-020542.
- Schachtsiek, J.; Hussain, T.; Azzouhri, K.; Kayser, O.; Stehle, F. Virus-Induced Gene Silencing (VIGS) in Cannabis Sativa L. Plant Methods2019, 15, 157, doi:10.1186/s13007-019-0542-5.
- Hung, Y.-H.; Slotkin, R.K. The Initiation of RNA Interference (RNAi) in Plants. Curr. Opin. Plant Biol.2021, 61, 102014, doi:10.1016/j.pbi.2021.102014.
- Hammond, S.M.; Bernstein, E.; Beach, D.; Hannon, G.J. An RNA-Directed Nuclease Mediates Post-Transcriptional Gene Silencing in Drosophila Cells. Nature2000, 404, 293–296, doi:10.1038/35005107.
- Burch‐Smith, T.M.; Anderson, J.C.; Martin, G.B.; Dinesh‐Kumar, S.P. Applications and Advantages of Virus‐induced Gene Silencing for Gene Function Studies in Plants. Plant J.2004, 39, 734–746, doi:10.1111/j.1365-313X.2004.02158.x.
- Chen, X.; Rechavi, O. Plant and Animal Small RNA Communications between Cells and Organisms. Nat. Rev. Mol. Cell Biol.2022, 23, 185–203, doi:10.1038/s41580-021-00425-y.
- Benedito, V.A.; Visser, P.B.; Angenent, G.C.; Krens, F.A. The Potential of Virus-Induced Gene Silencing for Speeding up Functional Characterization of Plant Genes. Genet. Mol. Res.2004.
- Chen, Y.G.; Hur, S. Cellular Origins of dsRNA, Their Recognition and Consequences. Nat. Rev. Mol. Cell Biol.2022, 23, 286–301, doi:10.1038/s41580-021-00430-1.
(4) Figure 3B, 4B, 5B
(Q) The top of bars is unusual. What is the small bars (I see three mini bars in the top of each bar)?
Response: Thanks for your comments. In our revised manuscript, we modified Figure 3B, 4B, 5B, The top of bars is the experiment was repeated three times and 15 or 20 leaves were used in each time, error bars represent standard errors. line 138, line 160, line 180.
(5) Figure legends for Fig. 3, 4, 5, and 6. A. CP protein virus detection.
(Q) I cannot understand this sentence. In these figure, the PCR-amplified DNA fragments are shown, and not for CP protein.
Response: Thank you for your constructive suggestion. In our revised manuscript, We correct the inaccurate words and try our best to make a scientific editing for Figure legends for Fig. 3, 4, 5, and 6. A, line 119 to line 122, line 146 to line 150, line 169 to line 171, line 188 to line 190.
(6) Figure legends for Fig. 3, 4, 5, and 6. B. CP protein detection rate.
(Q) The authors did not investigate any protein level of CP protein. They only detected the RNA level of TRV via detection of RNA encoding CP protein.
Response: Thank you for your constructive suggestion. In our revised manuscript, We correct the inaccurate words and try our best to make a scientific editing for Figure legends for Fig. 3, 4, 5, and 6. B, line 138, line 160, line 180, line 198.
(7) Text for the description of Fig. 3, 4, 5, and 7.
For example, the authors described the result of Figure 3A,3B as follows; the successful introduction of theTRV vector was confirmed base on the detection of the CP protein.
(Q) As already pointed out, the authors did not determine any protein level of CP protein. They only detected the level of RNA encoding CP protein.
Response: Thank you for your constructive suggestion. In our revised manuscript, We correct the inaccurate words and try our best to make a scientific editing, line 119 to line 122, line 146 to line 150, line 169 to line 171, line 188 to line 190.
(8) Figure legends for Fig. 3, 4, 5, and 6.
(Q) The details of each experiment were not shown. For example, in Fig. 3, the authors injected Agrobacterium suspension into the cotyledon two times. Then the Fig. 3D, they showed the leaf photograph. Which leaf was investigated? In addition, they mentioned that the TRV-HmPDS leaves showed photobleaching. However, the Fig. 3D (b) showed that the leaves in test sample was more greenish than the control sample. How do you judge the “photobleached” leaves ? Anyway the details of all experiments for Fig. 3, 4, 5 and 7 should be provided in more detail.
Response: Thank you for your constructive suggestion. In our revised manuscript, We correct the inaccurate words and try our best to make a scientific editing. The experiment deals with the cotyledons of JHK, observing the cotyledons as well as the newly leaves.
Injection efficiency was detected using PCR in the treated JHK seedlings, and leaves not injected with Agrobacterium (CK), pTRV1+pTRV2 and pTRV1+pTRV2-AmPDS were selected for PCR detection. The results showed that no band was seen in CK, while pTRV1+pTRV2 and pTRV1+pTRV2-AmPDS had a clear band at 270 bp (Figure 3A,B). This indicates that the back-of-blade injection method was successful in achieving infection of the leaves of JHK seedlings .. To assess further the efficiency of AmPDS silencing, qRT-PCR was performed to quantify AmPDS expression levels. The results indicated that AmPDS expression in the leaves of plants from the experimental group was significantly reduced compared with those from the control group, with AmPDS expression being reduced by nearly 60% (Figure 3C). Further research indicated that, at 2 days after the second injection, the cotyledons of JHK seedlings began to show photobleaching, whereas the negative-control plants exhibited no phenotypic changes, after 30 d, the leaves of those plants infected with pTRV1 + pTRV2-AmPDS Agrobacterium were restored to green (Figure 3D), the incidence of photobleaching was 54.4% (Figure 3E). Further revealed that the silencing of AmPDS significantly reduced the chlorophyll content in JHK leaves (Figure 3F).
(9) Text for the description of Fig. 3, 4, and 5.
For example, the authors described the result of these experiments as follows; JHK seedlings were inoculated via Agrobacterium infiltration. However the method for each experiment was different. So, the authors should provide unique sentences in each experiment. Do not repeat the similar sentences in the text!
Response: Thanks for your comments. In our revised manuscript, we modified these sentences for text for the description of Fig. 3, 4, and 5.
(10) Line 246. leaf abaxial injection method
(Q) The authors should unify the words. In line 124 and method paragraph, they use “Back-of blade injection”. However, they use here “leaf abaxial injection method” even though these two showed the same experiment.
Response: Thank you for your constructive suggestion. In our revised manuscript, we modified leaf abaxial injection method. We correct the inaccurate words and try our best to make a scientific editing, line 210.
(11) 4.6.1 Back-of-blade injection method
(Q) As mentioned in the previous query, authors injected the Agrobacterium solution into the abaxial side of cotyledon (same as back-of blade). However, in the method (4.6.1), authors mentioned that suspention was injected into the leaf side of the cotyledon. Usually, leaf side means the adaxial side and may be not abaxial side. Which was the true?
Response: Thank you for your constructive suggestion. In our revised manuscript, we modified this sentence. We correct the inaccurate words and try our best to make a scientific editing, line 317.
(12) Figure legends for Fig. 4 and 5.
The authors used “pressure side”
(Q) I cannot understand “pressure side”. It means the adaxial side.
Response: Thanks for your comments. The representation of “pressure side” means the adaxial side. We correct the inaccurate words and try our best to make a scientific editing, line 163 and line 183.
We tried our best to improve the manuscript and made some changes marked in red in revised paper which will not influence the content and framework of the paper. We appreciate for Editors and Reviewer’ warm work earnestly, and hope the correction will meet with approval.
Finally, we thank again for reviewers’ helpful comments and advices. We hope that the revisions adequately address the concerns raised during the review better meets the standards set by "Plants". We sincerely welcome comments for a further improvement of the paper. Thank you once again for your valuable guidance and support throughout the review process.
Sincerely,
Corresponding author: Li Cao
Affiliation: Agricultural College of Yanbian University
Email Address: lcao@ybu.edu.cn

Round 2
Reviewer 1 Report
Comments and Suggestions for Authors
The manuscript has been well revised. It can be published in the journal.
Author Response
Dear Editor,
We extend our heartfelt appreciation for your valuable time and dedicated effort expended in the review of our manuscript titled " Establishment of a Virus-induced Gene Silencing System in Hibiseu manihot L." (Manuscript ID: plants-3282314). We sincerely appreciate the constructive criticism provided and would like to express our gratitude for the insightful comments made by the reviewers. In accordance with the feedback from the editor and reviewers, we have diligently addressed the suggestions and concerns raised during the review process. We have carefully revised the manuscript, and the changes have been highlighted in red for easy reference in the revised version. Additionally, we have provided detailed responses to each reviewer's comments, which are presented below, along with the corresponding location of the revisions in the manuscript.
Reviewer 1
Dear Reviewer, I am writing to express our profound gratitude and appreciation for your meticulous review and valuable contributions throughout the process of evaluating our manuscript.
Your insightful comments and suggestions have been of utmost importance in enhancing the quality and clarity of our research work. Your in-depth understanding and critical analysis have not only helped us to refine the manuscript but have also deepened our own understanding of the subject matter. We are truly indebted to you for the time and effort you dedicated to this review.
Finally, we thank again for reviewers’ helpful comments and advices.
Sincerely,
Corresponding author: Li Cao
Affiliation: Agricultural College of Yanbian University
Email Address: lcao@ybu.edu.cn
Reviewer 2 Report
Comments and Suggestions for Authors
Reviewer 2
The revised manuscript still contains many points that should be addressed properly before publication.
comments (R means the author response. C means comment for the revised manuscript)
(1) Line 36. Hibiseu manihot belong to the Abelmoschus genus.
(R) we have corrected the “Hibiseu manihot” into “Abelmoschus manihot”.
(C) OK. I accepted.
(2) Line 36 to line 60
(R) we modified the part, line 40 to line 57.
(C) OK. I accepted.
(3) Line 62 to line 82
(R) we modified the part, line 59 to line 77.
(C1) Line 57-58, the 500 kDa complex is estimated from the Drosophila. I recommend the removal of ref 16 and the following sentence from the manuscript; “which consists of RNA and proteins, with a molecular weight of approximately 500 kDa [16]”.
(C2) Line 66. the analysis of gene function with greater speed and simlicity, Second, VIGS is time efficient. These two sentence is redundant. ‘greater speed’ and ‘time efficient’ are the same.
(C3) Line 67. ‘VIGS is broadly applicable”. This sentence is obscure. applicable to analysis of many genes? or applicable to analysis of broad range of plant species?
(4) Figure 3B, 4B, 5B
(R): we modified Figure 3B, 4B, 5B, The top of bars is the experiment was repeated three times and 15 or 20 leaves were used in each time, error bars represent standard errors. line 138, line 160, line 180.
(C): I cannot find the modification of Figure 3B, 4B, and 5B. I recommend that these figures can be revised according to the style appeared in Figure 6B.
(5) (6) Figure legends for Fig. 3, 4, 5, and 6.
(R): We try our best to make a scientific editing for Figure legends for Fig. 3, 4, 5, and 6. A, line 119 to line 122, line 146 to line 150, line 169 to line 171, line 188 to line 190.
(C) The revised figure legends are too primitive. For example, Figure 3 Silencing of AmPDS in a leaf of Abelmoschus manihot L. using the back-of-blade injection method. A, detection of viral RNA in injected leaves? (or detection of viral RNA in systemic leaves?). A fragment for the CP gene was amplified by RT-PCR (no description about the Fig.3A photograph!). a:,,, b:,,,, c:,,,,. B, Injection efficiency. this should mean that the authors determine the viral RNA in the injected leaves. If the authors determined the CP mRNA in the systemic leaves, in such case, the efficiency of viral systemic spreading. I cannot understand which is correct. Figure 3C, the heading is “HmPDS”. it should be “AmPDS”. Figure 3C, this data means the expression of PDS in injected leaves? (means cotyledon?) or upper leaves (means investigation of systemic inhibition of expression of PDS gene)?. Please describe more in detail for Fig. 3,4,5,and 6.
(7) and (8). Text and legend for the description of Fig. 3, 4, 5, and 7.
(R): We correct the inaccurate words and try our best to make a scientific editing, line 119 to line 122, line 146 to line 150, line 169 to line 171, line 188 to line 190.
(C)Fig. 3F, 4F, 5F. The data showed the VIGS effect on chlorophyll content. However, the proportion of VIGS are not 100% (Fig. 3E, 4E, 5E). This VIGS effects are obtained from all plants (including no phenotype) or plants showing VIGS ? If the latter case is correct, the number of plants should be provided.
(9) Text for the description of Fig. 3, 4, and 5.
(R): we modified these sentences for text for the description of Fig. 3, 4, and 5.
(C): I accepted.
(10) Line 246. leaf abaxial injection method
(R): we modified leaf abaxial injection method.
(C) I can still find ‘leaf abaxial” in line 219 and 255 of revised manuscript.
(11) 4.6.1 Back-of-blade injection method
(R): We correct the inaccurate words and try our best to make a scientific editing, line 317.
(C) I accepted.
(12) Figure legends for Fig. 4 and 5.
(R): The representation of “pressure side” means the adaxial side.
(C): I accepted.
Several sentences should be revised to make more clear.
Author Response
Dear Editor,
We extend our heartfelt appreciation for your valuable time and dedicated effort expended in the review of our manuscript titled " Establishment of a Virus-induced Gene Silencing System in Hibiseu manihot L." (Manuscript ID: plants-3282314). We sincerely appreciate the constructive criticism provided and would like to express our gratitude for the insightful comments made by the reviewers. In accordance with the feedback from the editor and reviewers, we have diligently addressed the suggestions and concerns raised during the review process. We have carefully revised the manuscript, and the changes have been highlighted in red for easy reference in the revised version. Additionally, we have provided detailed responses to each reviewer's comments, which are presented below, along with the corresponding location of the revisions in the manuscript.
Reviewer 2
The revised manuscript still contains many points that should be addressed properly before publication.
(1) Line 36. Hibiseu manihot belong to the Abelmoschus genus.
(R) we have corrected the “Hibiseu manihot” into “Abelmoschus manihot”.
(C) OK. I accepted.
(2) Line 36 to line 60
(R) we modified the part, line 40 to line 57.
(C) OK. I accepted.
(3) Line 62 to line 82
(R) we modified the part, line 59 to line 77.
(C1) Line 57-58, the 500 kDa complex is estimated from the Drosophila. I recommend the removal of ref 16 and the following sentence from the manuscript; “which consists of RNA and proteins, with a molecular weight of approximately 500 kDa [16]”.
Response: We appreciate your attention to detail and valuable feedback. In our revised manuscript, According to your suggestion, we modified the part, line 64.
In the maintenance stage, siRNA facilitates the formation of the RNA-induced silencing complex, the complex can be activated by unwound siRNAs, identify single-stranded siRNA sequences and degrade complementary transcriptional products.[16].
- Benedito, V.A.; Visser, P.B.; Angenent, G.C.; Krens, F.A. The Potential of Virus-Induced Gene Silencing for Speeding up Functional Characterization of Plant Genes. Genet. Mol. Res. GMR2004, 3, 323–341.
(C2) Line 66. the analysis of gene function with greater speed and simlicity, Second, VIGS is time efficient. These two sentence is redundant. ‘greater speed’ and ‘time efficient’ are the same.
Response: Thanks for your comments. In our revised manuscript, According to your suggestion, we modified the part, line 71.
First, VIGS is technically straightforward, requiring only Agrobacterium infiltration to induce gene silencing, without the need for complex genetic transformation steps, thereby allowing the analysis of gene function with greater speed and simplicity, with visible effects typically observed within 2–3 weeks.
(C3) Line 67. ‘VIGS is broadly applicable”. This sentence is obscure. applicable to analysis of many genes? or applicable to analysis of broad range of plant species?
Response: We were really sorry for our careless mistakes. Thanks for your reminder. In our revised manuscript, VIGS is applicable to analysis of broad range of plant species, line 74.
(4) Figure 3B, 4B, 5B
(R): we modified Figure 3B, 4B, 5B, The top of bars is the experiment was repeated three times and 15 or 20 leaves were used in each time, error bars represent standard errors. line 138, line 160, line 180.
(C): I cannot find the modification of Figure 3B, 4B, and 5B. I recommend that these figures can be revised according to the style appeared in Figure 6B.
Response: We sincerely thank the reviewer for careful reading. As suggested by the reviewer, in our revised manuscript, the original Figure B represents the percentage of the number of leaves in which bands can be detected relative to the total number of treated leaves. However, to avoid misunderstandings, we have decided to delete Figure B.
(5) (6) Figure legends for Fig. 3, 4, 5, and 6.
(R): We try our best to make a scientific editing for Figure legends for Fig. 3, 4, 5, and 6. A, line 119 to line 122, line 146 to line 150, line 169 to line 171, line 188 to line 190.
(C) The revised figure legends are too primitive. For example, Figure 3 Silencing of AmPDS in a leaf of Abelmoschus manihot L. using the back-of-blade injection method. A, detection of viral RNA in injected leaves? (or detection of viral RNA in systemic leaves?). A fragment for the CP gene was amplified by RT-PCR (no description about the Fig.3A photograph!). a:,,, b:,,,, c:,,,,.
Response: Thanks for the insightful comment. In our revised manuscript, we modified the part, line 138-152.
A,the target fragments of AmPDS were detected in new JHK leaves after injection; M means DNA marker; a present the PCR fragemenrt amplified from the leaves injected with pTRV1+pTRV2; b, DNA fragement from the leaves injected with pTRV1+pTRV2–AmPDS; c: leaves without injection (CK);
B, the phenotype of JHK leaves respectively injected with pTRV1+pTRV2, and pTRV1+pTRV2–AmPDS, up channel, 2 days after injection; bottom channel, 30 days after injection;
C, statistics of the number of phenotypes occurring and not occurring (P = 0.0132);
D,relative expression levels of AmPDS in JHK leaves respectively injected with pTRV1+pTRV2, and pTRV1+pTRV2–AmPDS. Three JHK leaves with phenotypes were selected as one biological replicate, and three biological replicates were set;
E, the chlorophyll content was detected in the leaves with phenotypes. Three JHK leaves with phenotypes were selected as one biological replicate, and three biological replicates were set.
All statistical analyses were performed with GraphPad Prism (v.9.0.0). Student’s t-test were used to assess differences between different treatments ( *: p < 0.05, **: p < 0.01).
B, Injection efficiency. this should mean that the authors determine the viral RNA in the injected leaves. If the authors determined the CP mRNA in the systemic leaves, in such case, the efficiency of viral systemic spreading. I cannot understand which is correct.
Response: Thank you for your constructive suggestion. In our revised manuscript, the original Figure B represents the percentage of the number of leaves in which bands can be detected relative to the total number of treated leaves, we have decided to delete the Figure B.
Figure 3C, the heading is “HmPDS”. it should be “AmPDS”. Figure 3C, this data means the expression of PDS in injected leaves? (means cotyledon?) or upper leaves (means investigation of systemic inhibition of expression of PDS gene)?. Please describe more in detail for Fig. 3,4,5,and 6.
Response: We sincerely appreciate your valuable input and suggestion. In our revised manuscript, this data means the expression of AmPDS in in the JHK new leaves with phenotypes, line 132.
To assess further the efficiency of AmPDS silencing, qRT-PCR was performed to quantify AmPDS expression levels. The results indicated that AmPDS expression in the JHK new leaves with phenotypes from the experimental group was significantly reduced compared the control group, and AmPDS expression being reduced by nearly 60% (Figure 3D).
(7) and (8). Text and legend for the description of Fig. 3, 4, 5, and 7.
(R): We correct the inaccurate words and try our best to make a scientific editing, line 119 to line 122, line 146 to line 150, line 169 to line 171, line 188 to line 190.
(C)Fig. 3F, 4F, 5F. The data showed the VIGS effect on chlorophyll content. However, the proportion of VIGS are not 100% (Fig. 3E, 4E, 5E). This VIGS effects are obtained from all plants (including no phenotype) or plants showing VIGS ? If the latter case is correct, the number of plants should be provided.
Response: Thank you very much for your careful comments. Figure 3E and Figure 6E means the chlorophyll content was detected in the leaves with phenotypes. Three JHK leaves with phenotypes were selected as one biological replicate, and three biological replicates were set. Figure 4E and Figure 5E means the chlorophyll content was detected in the leaves with phenotypes. Five JHK leaves with phenotypes were selected as one biological replicate, and three biological replicates were set, line 148,176,200 and 224.
(9) Text for the description of Fig. 3, 4, and 5.
(R): we modified these sentences for text for the description of Fig. 3, 4, and 5.
(C): I accepted.
(10) Line 246. leaf abaxial injection method
(R): we modified leaf abaxial injection method.
(C) I can still find ‘leaf abaxial” in line 219 and 255 of revised manuscript.
Response: We were really sorry for our careless mistakes. Thanks for your reminder, We have already revised “leaf abaxial” to “back-of-blade injection” as per the reviewer’s suggestion, line 243 and line 276.
(11) 4.6.1 Back-of-blade injection method
(R): We correct the inaccurate words and try our best to make a scientific editing, line 317.
(C) I accepted.
(12) Figure legends for Fig. 4 and 5.
(R): The representation of “pressure side” means the adaxial side.
(C): I accepted.
We tried our best to improve the manuscript and made some changes marked in red in revised paper which will not influence the content and framework of the paper. We appreciate for Editors and Reviewer’ warm work earnestly, and hope the correction will meet with approval.
Finally, we thank again for reviewers’ helpful comments and advices. We hope that the revisions adequately address the concerns raised during the review better meets the standards set by "Plants". We sincerely welcome comments for a further improvement of the paper. Thank you once again for your valuable guidance and support throughout the review process.
Sincerely,
Corresponding author: Li Cao
Affiliation: Agricultural College of Yanbian University
Email Address: lcao@ybu.edu.cn

Round 3
Reviewer 2 Report
Comments and Suggestions for Authors
The legend and corresponding text of Figure 3,4,5 and 6 were still primitive and also include many mistakes. The authors’ responses also include some contradiction. So extensive and careful revise is still required. In addtion, the revised sentences include several misspelled words!
(1) Figure 2 still include “pTRV2-HmPDS”. This should be “pTRV2-AmPDS”.
(2) Figure 3B, TRV-HmPDS should be TRV-AmPDS
(3) Figure 3D, HmPDS should be AmPDS
(4) Figure 4B, TRV-HmPDS should be TRV-AmPDS
(5) Figure 4D, HmPDS should be AmPDS
(6) Figure 5B, TRV-HmPDS should be TRV-AmPDS
(7) Figure 5D, HmPDS should be AmPDS
(8) Figure 6B, HmPDS should be AmPDS
(9) Figure 6D, HmPDS should be AmPDS
(10) Legend for Figure 3A, 4A, 5A and 6A, What is “target fragment”? The revised sentences is getting worse! The fragment would be the RT-PCR product of CP mRNA. In the revised manuscript, the description about this is disappeared!
(11) About the CP mRNA detection, in the Materials and Methods, there is no primers for CP mRNA detection.
(12) Figure 3B, the authors described as follows; phenotype of JHK leaves injected (line 144). However, in Materials and methods, the authors mentioned as follows: bacteria solution was injected into cotyledon (line 340). The photo of Figure 3B apparently shows the true leaves, not cotyledon. I speculate that the authors intention would be : phenotype of JHK leaves of the plants injected with TRV and TRV-HmPDS into cotyledons.
The “injected leaves” means the leaves directly injected with Agrobacterium, and any readers cannot speculated the true leaves of plants which cotyledon was injected with Agrobacterium. So the authors should carefully revise this point throughout the Figure 3, 4, 5 and 6.
(13) Text, line 123-124, this sentence is not correct! From the revised sentence, I have to understand as follows: The CP mRNA was detected in the JHK seedlings of CK (control plant). It is totally mistake.
Comments on the Quality of English LanguageSome parts of revised manuscript is getting worse! Careful revise is still required.
Author Response
Dear Editor,
We extend our heartfelt appreciation for your valuable time and dedicated effort expended in the review of our manuscript titled " Establishment of a Virus-induced Gene Silencing System in Hibiseu manihot L." (Manuscript ID: plants-3282314). We sincerely appreciate the constructive criticism provided and would like to express our gratitude for the insightful comments made by the reviewers. In accordance with the feedback from the editor and reviewers, we have diligently addressed the suggestions and concerns raised during the review process. We have carefully revised the manuscript, and the changes have been highlighted in red for easy reference in the revised version. Additionally, we have provided detailed responses to each reviewer's comments, which are presented below, along with the corresponding location of the revisions in the manuscript.
Reviewer 2
Comments and Suggestions for Authors
The legend and corresponding text of Figure 3,4,5 and 6 were still primitive and also include many mistakes. The authors’ responses also include some contradiction. So extensive and careful revise is still required. In addtion, the revised sentences include several misspelled words!
- Figure 2 still include “pTRV2-HmPDS”. This should be “pTRV2-AmPDS”.
Response: We were really sorry for our careless mistakes. Thanks for your reminder. In our revised manuscript, We have modified pTRV2-HmPDS in Figure 2 to pTRV2-AmPDS.
(2) Figure 3B, TRV-HmPDS should be TRV-AmPDS
Response: We were really sorry for our careless mistakes. Thanks for your careful checks. In our revised manuscript, We have modified pTRV2-HmPDS in Figure 3B to pTRV2-AmPDS.
(3) Figure 3D, HmPDS should be AmPDS
Response: We were really sorry for our careless mistakes. Thanks for the insightful comment. In our revised manuscript, We have modified pTRV2-HmPDS in Figure 3D to pTRV2-AmPDS.
(4) Figure 4B, TRV-HmPDS should be TRV-AmPDS
Response: We were really sorry for our careless mistakes. Thanks for your reminder. In our revised manuscript, We have modified pTRV2-HmPDS in Figure 4B to pTRV2-AmPDS.
(5) Figure 4D, HmPDS should be AmPDS
Response: We were really sorry for our careless mistakes. We sincerely thank the reviewer for careful reading. In our revised manuscript, We have modified pTRV2-HmPDS in Figure 4D to pTRV2-AmPDS.
(6) Figure 5B, TRV-HmPDS should be TRV-AmPDS
Response: We were really sorry for our careless mistakes. Thanks for your reminder. In our revised manuscript, We have modified pTRV2-HmPDS in Figure 5B to pTRV2-AmPDS.
(7) Figure 5D, HmPDS should be AmPDS
Response: We were really sorry for our careless mistakes. Thanks for your careful checks. In our revised manuscript, We have modified pTRV2-HmPDS in Figure 5D to pTRV2-AmPDS.
(8) Figure 6B, HmPDS should be AmPDS
Response: We were really sorry for our careless mistakes. Thanks for your comments. In our revised manuscript, We have modified pTRV2-HmPDS in Figure 6B to pTRV2-AmPDS.
(9) Figure 6D, HmPDS should be AmPDS
Response: We were really sorry for our careless mistakes. We sincerely thank the reviewer for careful reading. In our revised manuscript, We have modified pTRV2-HmPDS in Figure 6D to pTRV2-AmPDS.
(10) Legend for Figure 3A, 4A, 5A and 6A, What is “target fragment”? The revised sentences is getting worse! The fragment would be the RT-PCR product of CP mRNA. In the revised manuscript, the description about this is disappeared!
Response: We sincerely appreciate your valuable input and suggestion. In our revised manuscript, we have revised the inaccurate expressions and re-described this part of the content, line 145-147.
Fig. 3 the evaluation of back-of-blade injection method for the silencing of AmPDS
A, the detection of mRNA encoded CP protein using PCR assay in different group; M, DNA marker; a, PCR product from samples injected with pTRV1+pTRV2; b, PCR product from samples injected with pTRV1+pTRV2–AmPDS; c, PCR product from samples without injection.
- About the CP mRNA detection, in the Materials and Methods, there is no primers for CP mRNA detection.
Response: Thanks for the insightful comment. We have provided the primers for CP mRNA detection.
Table 4 Primer sequence
Primer |
Sequence |
||
CP mRNA |
Forward |
cctgctgacttgatggacga |
|
Reverse |
ccagtgttcgccttggtag |
(12) Figure 3B, the authors described as follows; phenotype of JHK leaves injected (line 144). However, in Materials and methods, the authors mentioned as follows: bacteria solution was injected into cotyledon (line 340). The photo of Figure 3B apparently shows the true leaves, not cotyledon. I speculate that the authors intention would be : phenotype of JHK leaves of the plants injected with TRV and TRV-HmPDS into cotyledons.
The “injected leaves” means the leaves directly injected with Agrobacterium, and any readers cannot speculated the true leaves of plants which cotyledon was injected with Agrobacterium. So the authors should carefully revise this point throughout the Figure 3, 4, 5 and 6.
Response: Thank you for your constructive suggestion. in present research, we detected the CP mRNA both in the cotyledons and true leaves for exploring that whether TRV system was transformed successfully into JHK and further visualizing silencing efficiency. On the surface of true leaves, the photobleaching phenomenon was observed, accounting 54.4%. However, the phenomenon disappeared after 30 days.
Similarly, the experimental results reported in the following literature are consistent with those in our article. The following is a brief description of this literature in response to the reviewers' questions: The jujube seedlings were injected with a syringe. Using a needle to make a slight scar on the back of the cotyledon, a 1 mL sterile syringe without the needle was then used to align the wound and inject the mixed bacteria solution. The infected jujube seedlings were first dark-treated for 24 h, and then cultured in a culture chamber (light 16 h/dark 8 h, 23 ℃, 60% of humidity). At 15d after injection, the target fragment was detected successfully in newly leaves of jujube by PCR (Figure 2C), suggesting that Agrobacteria carrying a pTRV2-ZjCLA recombination were transferred on jujube with the assistance of pTRV1. Figures A and B are phenotypic diagrams that appeared on the true leaves after the cotyledons were treated.
Zhang, Yao et al. “Virus-Induced Gene Silencing (VIGS) in Chinese Jujube.” Plants (Basel, Switzerland) vol. 12,11 2115. 26 May. 2023, doi:10.3390/plants12112115.
(13) Text, line 123-124, this sentence is not correct! From the revised sentence, I have to understand as follows: The CP mRNA was detected in the JHK seedlings of CK (control plant). It is totally mistake.
Response: We were really sorry for our careless mistakes. Thanks for your reminder. In our revised manuscript, we have modified this part of the content, line 123-127.
Firstly, we collected the samples of different groups and the PCR assay was employed to detect the mRNA which encoded CP protein, a symbolical product from TRV2 vector. The results showed that there was a clear band in pTRV1 + pTRV2 and pTRV1 + pTRV2 - AmPDS group while no band in CK group (Fig. 3A), illustrating that the TRV system was successfully transformed into JHK via back-of-blade injection method.
We tried our best to improve the manuscript and made some changes marked in red in revised paper which will not influence the content and framework of the paper. We appreciate for Editors and Reviewer’ warm work earnestly, and hope the correction will meet with approval.
Finally, we thank again for reviewers’ helpful comments and advices. We hope that the revisions adequately address the concerns raised during the review better meets the standards set by "Plants". We sincerely welcome comments for a further improvement of the paper. Thank you once again for your valuable guidance and support throughout the review process.
Sincerely,
Corresponding author: Li Cao
Affiliation: Agricultural College of Yanbian University
Email Address: lcao@ybu.edu.cn

Round 4
Reviewer 2 Report
Comments and Suggestions for Authors
There are only minor comments.
(1) Line 1, title: Abelmoschus manihot should be italic.
(2) Several species name (manihot) are shown in “Manihot” in Funding section and Reference section. In general, species name shoud be shown in small letters, as ‘manihot”. Check before publication.
(3) Line 177, “vacuum infiltrated with” may be “vacuum-infiltrated with”
(4) Table 4, primer sequences should be shown in upper case letters like Table 2.
Author Response
Dear Editor,
We extend our heartfelt appreciation for your valuable time and dedicated effort expended in the review of our manuscript titled " Establishment of a Virus-induced Gene Silencing System in Hibiseu manihot L." (Manuscript ID: plants-3282314). We sincerely appreciate the constructive criticism provided and would like to express our gratitude for the insightful comments made by the reviewers. In accordance with the feedback from the editor and reviewers, we have diligently addressed the suggestions and concerns raised during the review process. We have carefully revised the manuscript, and the changes have been highlighted in red for easy reference in the revised version. Additionally, we have provided detailed responses to each reviewer's comments, which are presented below, along with the corresponding location of the revisions in the manuscript.
Reviewer 2
Comments and Suggestions for Authors
There are only minor comments.
(1) Line 1, title: Abelmoschus manihot should be italic.
Response: Thank you for your constructive suggestion, we feel sorry for our carelessness. In our revised manuscript, we have corrected the font formatting of “Abelmoschus manihot ”in the revised manuscript as per the reviewer’s suggestion, line 1.
Establishment of a Virus-induced Gene Silencing System in Abelmoschus manihot L.
(2) Several species name (manihot) are shown in “Manihot” in Funding section and Reference section. In general, species name shoud be shown in small letters, as ‘manihot”. Check before publication.
Response: Thanks for your careful checks. We are sorry for our carelessness. Based on your comments, we modified the “manihot” in our entire revised manuscript.
(3) Line 177, “vacuum infiltrated with” may be “vacuum-infiltrated with”
Response: Thank you for your constructive suggestion. According to the review, we correct the inaccurate words and try our best to make a scientific editing, line 176-186.
(4)Table 4, primer sequences should be shown in upper case letters like Table 2.
Response: We appreciate your attention to detail and valuable feedback. According to your suggestion, we modified Table 4.
Table 4 Primer sequences
Primer |
Sequence (5¢ to 3¢) |
||
TRV CP mRNA |
Forward |
CCTGCTGACTTGATGGACGA |
|
Reverse |
CCAGTGTTCGCCTTGGTAG |
We tried our best to improve the manuscript and made some changes marked in red in revised paper which will not influence the content and framework of the paper. We appreciate for Editors and Reviewer’ warm work earnestly, and hope the correction will meet with approval.
Finally, we thank again for reviewers’ helpful comments and advices. We hope that the revisions adequately address the concerns raised during the review better meets the standards set by "Plants". We sincerely welcome comments for a further improvement of the paper. Thank you once again for your valuable guidance and support throughout the review process.
Sincerely,
Corresponding author: Li Cao
Affiliation: Agricultural College of Yanbian University
Email Address: lcao@ybu.edu.cn
